# Development and validation of FootNet; a new kinematic algorithm to improve foot-strike and toe-off detection in treadmill running

Adrian Rivadulla[1]*, Xi Chen[2], Gillian Weir[3], Dario Cazzola[1], Grant Trewartha[1], Joseph Hamill[3], Ezio Preatoni[1]

**1** Department for Health, University of Bath, Bath, United Kingdom, **2** Department of Computer Science, University of Bath, Bath, United Kingdom, **3** Department of Kinesiology, University of Massachusetts Amherst, Amherst, MA, United States of America

* A.Rodriguez.Rivadulla@bath.ac.uk

## Abstract

The accurate detection of foot-strike and toe-off is often critical in the assessment of running biomechanics. The gold standard method for step event detection requires force data which are not always available. Although kinematics-based algorithms can also be used, their accuracy and generalisability are limited, often requiring corrections for speed or foot-strike pattern. The purpose of this study was to develop FootNet, a novel kinematics and deep learning-based algorithm for the detection of step events in treadmill running. Five treadmill running datasets were gathered and processed to obtain segment and joint kinematics, and to identify the contact phase within each gait cycle using force data. The proposed algorithm is based on a long short-term memory recurrent neural network and takes the distal tibia anteroposterior velocity, ankle dorsiflexion/plantar flexion angle and the anteroposterior and vertical velocities of the foot centre of mass as input features to predict the contact phase within a given gait cycle. The chosen model architecture underwent 5-fold cross-validation and the final model was tested in a subset of participants from each dataset (30%). Non-parametric Bland-Altman analyses (bias and [95% limits of agreement]) and root mean squared error (RMSE) were used to compare FootNet against the force data step event detection method. The association between detection errors and running speed, foot-strike angle and incline were also investigated. FootNet outperformed previously published algorithms (foot-strike bias = 0 [–10, 7] ms, RMSE = 5 ms; toe-off bias = 0 [–10, 10] ms, RMSE = 6 ms; and contact time bias = 0 [–15, 15] ms, RMSE = 8 ms) and proved robust to different running speeds, foot-strike angles and inclines. We have made FootNet's source code publicly available for step event detection in treadmill running when force data are not available.

**Data Availability Statement:** All processed data for model training and testing are available in the University of Bath Research Data Archive (DOI: 10.

15125/BATH-00965). FootNet's source code and additional scripts to replicate model validation and testing can be found in GitHub (https://github.com/adrianrivadulla/FootNet).

**Funding:** This work was part of a PhD project match funded by the University of Bath and Nurvv, Ltd.

**Competing interests:** The authors have declared that no competing interests exist.

## Introduction

Running is one of the most popular sports around the world [1, 2] and one of the most studied actions within human movement research. Biomechanical analyses of running technique typically involve the identification of gait cycles (i.e. strides) and rely on the accurate detection of foot-strike and toe-off within each cycle. Foot-strike and toe-off are also used to compute basic running metrics associated with performance and injury, two of the main foci of running biomechanics research and consumer-based running technology. The current "gold standard" to identify foot-strike and toe-off for both overground and treadmill running is the onset and offset of the vertical ground reaction force (vGRF) that exceeds a particular force magnitude. However, force plates are typically limited to small areas in overground running studies making it difficult to collect several consecutive steps and force-instrumented treadmills are costly and only available in a few laboratories. As an alternative, conventional treadmills are often used in running kinematics research and gait clinics, requiring the estimation of step events without force data, which has historically proved challenging.

There are several algorithms that use kinematic quantities as input for step detection in treadmill running within the current literature [3–5]. Researchers have explored different variables to detect contact events including anatomical landmark trajectories (mainly on the foot) and their derivatives, and segment/joint kinematics [5, 6]. Algorithms based on landmark trajectories are simpler but typically require markers to be affixed to the shoe or foot in highly deformable areas (e.g. fifth metatarsal head, hallux). The repeated impacts and metatarsal joint dorsiflexion during late stance can compromise marker fixation especially in longer trials, leading to losing a marker or introducing noise into the marker trajectory signal. Noise can be further amplified when differentiating the signals, potentially affecting the accuracy and reliability of methods using high order derivatives. Algorithms using segment or joint kinematics as input may be more robust to noise in individual marker trajectories but they can be sensitive to errors in marker placement, which affect the computation of segment and joint kinematics [7–9].

Despite the multiple studies in kinematics-based step event detection, current algorithms require specific marker configurations, can be affected by running speed or foot-strike patterns [4] and validations have only been conducted in a single laboratory, potentially limiting their generalisability [3]. Furthermore, algorithm assessment has predominantly been limited to accuracy comparisons against a "gold standard" or other algorithms, but there has been little discussion about the impact that errors in the estimation of step events may have on the quantification and assessment of kinematic variables commonly studied in running biomechanics. Osis et al. [10] showed that typically accepted errors of 20 ms in the detection of foot-strike can imply a change in knee flexion angle of up to 7° at relatively low running speeds (2.65 ± 0.22 m/s). With joint angular velocity expected to increase at higher speeds, the impact of a 20 ms error on a given variable can become larger, requiring careful consideration. This highlights: a) the need for a more comprehensive understanding of the sensitivity of commonly studied kinematic variables at foot-strike and toe-off to errors in step detection; and b) the room for improvement in the accuracy, reliability and generalisability of step detection algorithms based on kinematics.

Recently, deep learning methods have provided solutions to complex pattern recognition and signal processing problems [11]. A deep learning model is a hierarchical network of functions organised in several interconnected layers [12]. Function parameters (i.e. weights and bias) are optimised in the training process to map a set of input features to desired output labels. Amongst the different network architectures, recurrent neural networks (RNN), and more specifically long short-term memory neural networks (LSTM) [13] are particularly well

suited for supervised learning tasks [14] on sequential data such as the kinematic and kinetic time-series typically studied in biomechanics. LSTM networks have proved successful in the identification of events in audio signals [15] and the classification of electrocardiogram signals [16]. In recent studies [17], LSTM has also been used in biomechanics to detect step events in children with gait disorders. Numerical results showed that LSTM outperformed existing algorithms, which makes it a promising candidate for foot-strike and toe-off event detection in treadmill running.

Therefore, the purpose of this study was to develop and evaluate FootNet, a novel kinematics and deep learning-based algorithm for foot-strike and toe-off event detection during treadmill running. The algorithm is based on an LSTM network and has been trained and tested using data collected under different running conditions (e.g. speeds, foot-strikes, inclines) and laboratories. We also investigated the sensitivity of different kinematic variables to errors in step detection.

## Materials & methods

### Data collection

We gathered five treadmill running datasets including lower limb kinematics and GRFs. These data were collected in three independent laboratories and present a wide variety of participant characteristics, testing protocols and equipment (Table 1):

- *Foot-strikes dataset* is an open access dataset [18] including 28 experienced athletes with rearfoot, midfoot and forefoot strike patterns. The testing protocol was approved by the Universidade Federal do ABC Ethics committee and written informed consent was obtained from each participant prior to participation in the study. Participants were captured using a 12-camera motion capture system (Motion Analysis, Santa Rosa, CA, USA) collecting at 150 Hz, and a dual-belt instrumented treadmill (FIT, Bertec, Columbus, OH, USA) collecting at 300 Hz synchronised through Cortex 6.0 software (Motion Analysis, Santa Rosa, CA, USA).

- *Inclines dataset* is another open access dataset [19] including 10 recreational participants running at different speeds on positive and negative gradients. Ethical approval for the testing protocol was granted by the Institutional Review Board at Vanderbilt University and all participants gave written informed consent prior to completing the test. Participants were captured using a 10-camera motion capture system (Vicon Motion Systems, Oxford, UK) collecting at 200 Hz and a dual-belt instrumented treadmill (Bertec Corporation, Columbus, Ohio, US) collecting at 1000 Hz synchronised through Vicon Nexus 2.9 software (Vicon Motion Systems, Oxford, UK).

- *Speed dataset* included 15 recreational participants running at a wide range of speeds (i.e. 2.5–5.0 m/s at 0.5 m/s increments).

- *Footwear dataset* included 11 long distance runners running at a fixed speed and their preferred speed under different athletic footwear conditions.

- *Prolonged run dataset* [20] included 16 recreational runners running at their preferred speed under two athletic footwear conditions (i.e. neutral and stability), running for two bouts of 21 minutes with a 2-minute break in between during which data were recorded every 5 minutes.

Approval for the studies relating to the Speed, Footwear and Fatigue datasets was gained from the institutional review board at the University of Massachusetts and written informed

**Table 1. Participant demographics and testing conditions.**

| Dataset | Pts | Age (yrs) | Height (m) | Body mass (kg) | Strike pattern | Speed (m/s) | Incline ( ) | Shoes |
|---------|-----|-----------|------------|----------------|----------------|-------------|-------------|-------|
| *Foot-strikes* | 28 M | 35± | 1.76± | 69.6± | 14 RF | 2.5–4.5[1] | 0 | Preferred |
| | | 7 | 0.07 | 7.7 | 5 MF | | | |
| | | | | | 9 FF | | | |
| *Inclines* | 5 M | 24± | 1.70± | 66.7± | 6 RF | 2.6–4 | -9–9 | Preferred |
| | 5 F | 3 | 0.10 | 6.4 | 3 MF | | | |
| | | | | | 1 FF | | | |
| *Speed* | 15 M | 22 ± 2 | 1.76 ± 0.1 | 74 ± 7 | 13 RF | 2.5–5[2] | 0 | Racing |
| | | | | | 1 MF | | | Flat |
| | | | | | 1 FF | | | |
| *Footwear* | 6 M | 30± | 1.67± | 57.5± | 7 RF | 3.35 | 0 | 5 neutral models |
| | 5 F | 8 | 0.04 | 3.5 | 1 MF | 4± | | |
| | | | | | 3 FF | 0.5[3] | | |
| *Prolonged* | 16 M | 24± | 1.78± | 70.1± | 16 RF | 3.2± | 0 | Neutral |
| | | 4 | 0.05 | 8 | 2 FF | 0.4[3] | | Stability |

Pts: Participants, M: Males, F: Females, RF: Rearfoot strikers, MF: Midfoot strikers, FF: Forefoot strikers. Foot-strike patterns for the Inclines dataset were identified in the 0 incline condition.

[1] Participants ran at speeds ranging between 2.5 and 4.5 m/s in increments of 1 m/s.

[2] Participants ran at speeds ranging between 2.5 and 5 m/s in increments of 0.5 m/s.

[3] Participants ran at their preferred speed.

consent for all participants was obtained. These three datasets were collected using an 8-camera motion capture system (Qualysis, Inc., Gothenburg, Sweden) collecting at 200 Hz and an instrumented treadmill (Treadmetrix, Park City, UT) collecting at 2000 Hz synchronised through Qualysis Track Manager (Qualysis, Inc., Gothenburg, Sweden).

For each dataset, three-dimensional marker trajectories were low-pass filtered (Butterworth, 4th order, zero lag) with a cut off frequency of 10 Hz [18]. Filtered marker trajectories were used to estimate the position and orientation of body segments using a six degrees-of-freedom modelling approach in Visual3D (Visual 3D™, C-Motion, USA). The anatomical definitions of the shank and a single-segment foot were kept consistent for every dataset but different tracking markers for the shank and foot were selected for each dataset (Fig 1). This allowed us to increase the variability within the data during algorithm development and to validate the model for different tracking marker sets. Ankle joint angles were calculated using the Cardan sequence flex/extension, abd/adduction and int/external rotation. Full trials were firstly divided in cycles using the highest position of the foot centre of mass (COM) and the contact phase of each cycle was identified within the raw vGRF. Although the onset and offset

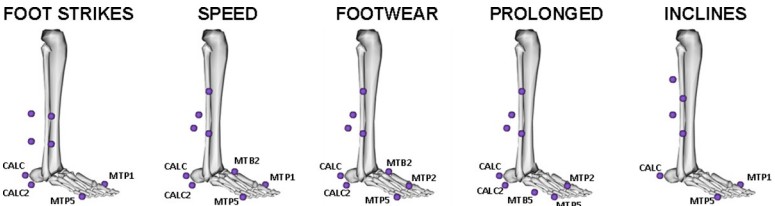

**Fig 1. Tracking markers used for each dataset.** The shank was tracked using clusters at different positions and different marker configurations. CALC: calcaneus, CALC2: lateral calcaneus, MTB5: 5th metatarsal base, MPT5: 5th metatarsal head, MTB2: 2nd metatarsal base, MTP2: 2nd metatarsal head, MTP1: 1st metatarsal head.

of vGRF is regarded as the "gold standard" method for step event detection, it must be noted that instrumented treadmills are not as reliable as floor-embedded force plates, especially at low loads. This is due to the bending stiffness of the treadmill deck and the moving belt. We noticed aberrant plateaus/bumps in the vGRF signal in some cycles when approaching toe-off, where a steep decreasing vGRF is expected. Therefore, we systematically assessed vGRF signals and discarded those cycles exhibiting a positive loading rate ≥200 N/s (i.e. 1 N per frame, if sampling at 200 Hz) in the interval where vGRF dropped from 100 N to 50 N (i.e. 1677 out of 36212 cycles, 4.63%). The contact phase was defined in the accepted cycles using a 50 N threshold (non-contact: vGRF < 50 N; contact: vGRF ≥ 50 N). The first and last frame of contact within each cycle were identified and the closest frames at motion capture sampling frequency were selected as foot-strike and toe-off. A label vector (i.e. 0 non-contact and 1 contact for each time point at motion capture sampling frequency) was created for each cycle and used as "ground truth" for algorithm development.

## Step detection algorithm architecture and development

This study approached the detection of foot-strike and toe-off as a binary classification problem. FootNet aims to predict the contact and non-contact phases within a given cycle based on a set of kinematic input features. Foot-strike and toe-off can then be simply identified by finding the start and end of the contact phase. Shank and foot COM, proximal and distal end velocities and accelerations as well as ankle joint angles, angular velocities and angular accelerations were visually inspected for salient attributes within the gait cycle (e.g. peaks or troughs) that could facilitate the identification of foot-strike and toe-off. We avoided displacements as they depend on the definition of the global coordinate system and high order derivatives to prevent noise amplification. The anteroposterior velocity of the distal tibia, ankle dorsi/plantarflexion angle and the anteroposterior and vertical velocity of the foot COM, were selected as the best input feature candidates for contact phase prediction after initial experimentation.

Consider defining the input data sequences as $X = [X_1, X_2, . . .,X_n]^T$, where $n$ is the number of input features ($n = 4$), and $X_n$ is a sequence of input values $\{x_t\}$ with length $t = 200$. The output sequence (or output label) is defined as $Y = [y_1, y_2, . . .,y_t]^T$, where $y_t$ is a binary value. The proposed FootNet algorithm is developed based on RNN architecture with LSTM units, which are known to mitigate the gradient vanishing problem faced by traditional RNNs when doing back-propagation through time in long input sequences [13]. An LSTM unit at timepoint $t$ (Fig 2) is characterised by its memory cell or *cell state*, $c_t$, which controls what gradient information is maintained or discarded, and its *hidden state*, $h_t$, which is the output of the LSTM unit. The cell state acts as internal memory of the LSTM unit, carrying information about previous time along the entire sequence. In a typical LSTM unit, the cell state is controlled by three gating functions: forget gate $f_t$, input gate $i_t$ and output gate $o_t$, which can be computed by three separate but similar functions:

$$\begin{bmatrix} f_t \\ i_t \\ o_t \end{bmatrix} = \begin{pmatrix} \sigma \\ \sigma \\ \sigma \end{pmatrix} \widehat{W}_t[h_{t-1}, x_t]^T \tag{1}$$

where matrix $\widehat{W}_t$ contains three vectors representing the weight of the three gates, respectively. $\begin{pmatrix} \sigma \\ \sigma \\ \sigma \end{pmatrix}$ denotes the three separate Sigmoid functions, the output of which ranges between

# LSTM unit

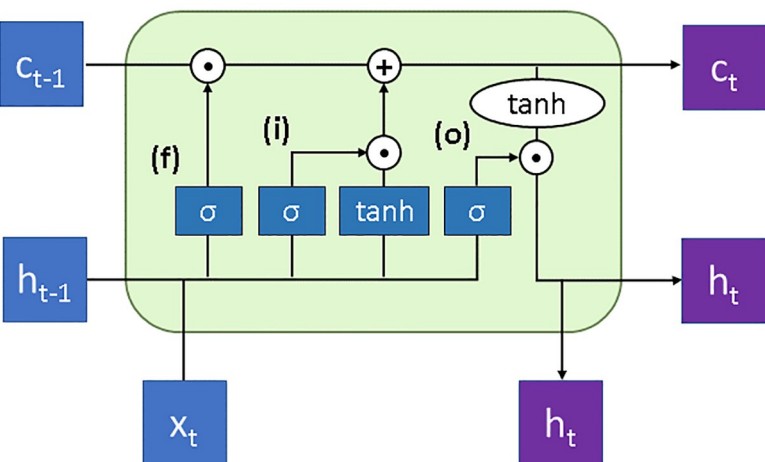

**Fig 2. LSTM unit.** Graphical representation of an LSTM unit at timepoint t, where c and h represent the cell state and hidden state respectively, x represents the input features at the current timepoint, and f, i o are the forget, input and output gates controlling the information passed on to the cell state.

0 and 1. The memory cell $c_t$, is then calculated by:

$$c_t = f_t \odot c_{t-1} + i_t \odot \tanh(\widehat{w}_c[h_{t-1}, x_t]^T) \tag{2}$$

where $\odot$ represents the element-wise multiplication operation, vector $\widehat{w}_c$ denotes the weight of the memory cell, and tanh(.) is the standard hyperbolic tangent activation function. The output of the LSTM hidden node is computed by:

$$h_t = o_t \odot \tanh c_t \tag{3}$$

The proposed FootNet model uses a variant of LSTM units called bidirectional LSTM (Bi-LSTM), aiming to efficiently make use of both the forward state and backward state features [21] in our step detection problem. Specifically, for each unit going forward in time, there is a parallel unit that takes the same sequence as input but flipped. This allows the network to not only learn relationships between previous time steps but also the following ones.

As shown in Fig 3, the proposed model consists of three hidden layers including two bidirectional LSTM layers with 400 units each and a dense layer with 200 nodes and rectified linear unit (ReLU) as activation function. The output layer has only one node with a Sigmoid activation function. Model parameters were optimised to minimise the binary cross-entropy loss between the predicted $\widehat{y}_t$ and ground truth label vectors $y_t$ using Adam optimiser [22]. Training data $\{X, Y\}$ were fed in mini-batches of 200 training cycles and model parameters were optimised after every mini-batch. Hyperparameter combinations, including the number of hidden layers and units/nodes per layer, activation function, optimiser and mini-batch size were determined by empirical experimentation using a grid search approach. Dropout layers (with dropout ratio = 0.5) [23] were used after the LSTM layers and the fully connected layer to mitigate overfitting during model training. The number of training epochs was limited to 100 and network training was terminated if the reported validation accuracy converged (i.e. no validation improvement in 10 epochs). To maximise the variability within the training data, we mixed participants from different datasets. FootNet underwent 5-fold cross-validation [24] as shown in

## FootNet architecture

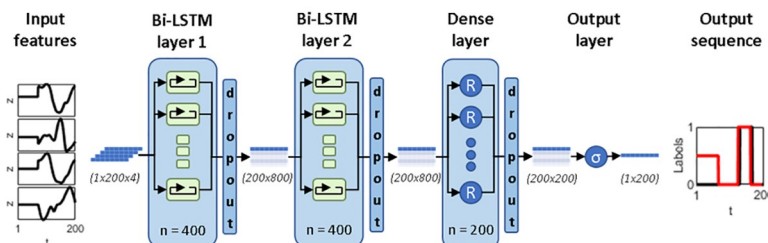

**Fig 3. FootNet architecture.** FootNet takes the distal tibia anteroposterior velocity, ankle plantar/dorsiflexion angle and the foot COM anteroposterior and vertical velocities as input and produces a sequence of probabilities of non-contact (0) and contact (1). Note that the input features are first standardised (z scores), 0-padded at the start of the sequence to a standard length of 200 data-points and concatenated into a 1x200x4 array where 1 is the number of cycles, 200 is the number of timepoints and 4 is the number of input features. Zero padding was added to batch the training cases and 200 was selected as a sufficient number of timepoints to accommodate a stride cycle at any of the tested speeds. The padding is ignored during training, hence the 0.5 values at the beginning of the output sequence.

Fig 4. Data were standardised to z-scores using the mean and standard deviation from the training set as scaling factors for the validation set during cross-validation and for the testing set during testing. This prevented any information leakage from the validation and testing sets.

Neural network architecture and training routines were developed in Python (Python 3.7, Python Software Foundation, Wilmington, DE, USA) using custom scripts and the Keras library (Keras 2.3.0, https://keras.io) within the TensorFlow machine learning framework (TensorFlow 2.3, https://www.tensorflow.org). Network training was performed on a GPU (NVIDIA Tesla P100, 16 GB RAM) operated by a computer (Intel (R) Xeon (R) processor with two cores @2.3 GHz, operating system: Linux 4.19.104+, Ubuntu distribution: Ubuntu 18.04).

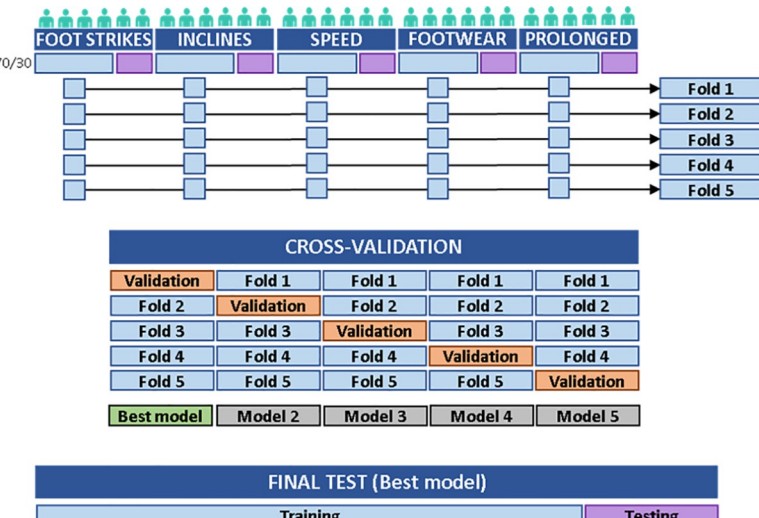

**Fig 4. Data flow.** From each dataset, 30% of the participants were extracted, concatenated and left aside as test set for model testing (22 participants, 8712 cycles). The remaining 70% of the participants were divided in five groups and concatenated in five folds with a representative number of participants from each dataset (10 to 14 participants each, 4999 ± 842 cycles in each fold). These five folds were used for cross-validation, whereby five models (i.e. sets of parameters) were developed using four folds for training and the remaining fold for validation, with a different validation fold for each model. The best performing model was selected, retrained using the five folds altogether and then tested on the test set.

## Step detection algorithm assessment

Mean (± standard deviation) cross-validation accuracy was calculated and the best performing model (i.e. set of parameters) was selected and trained again using the entire training set (i.e. the concatenated five folds). The performance of this model was assessed on the test set. Model accuracy on the test set is reported for contact and non-contact classification. A Bland-Altman analysis [25] was conducted to assess the agreement between our algorithm and the "gold standard" method for the detection of foot-strike, toe-off and contact times. Error distributions were tested for normality using the Kolmogorov-Smirnov test. Due to the non-normal distribution of errors in foot-strike (D = 0.495, p < 0.001), toe-off (D = 0.494, p < 0.001) and contact time (D = 0.491, p < 0.001) detection, bias and 95% limits of agreement (95LA) were estimated non-parametrically as median (bias) and 2.5th and 97.5th percentile (lower and upper 95% limits of agreement) [26]. Root mean squared error (RMSE) and Pearson's correlation coefficient (r) were calculated to assess the linear association between errors in foot-strike, toe-off and contact time estimation, and running speed, foot angle and incline, respectively. Correlation coefficients were classified as trivial (r < 0.1), small (r ≥ 0.1 and < 0.3), moderate (r ≥ 0.3 and < 0.5) and large (r ≥ 0.5) [27]. Linear regression coefficients and coefficient of determination ($r^2$) were calculated for non-trivial correlations. The level of significance for normality tests and correlation coefficients was set at α = 0.05.

## Sensitivity analysis

We also studied the sensitivity of hip, knee and ankle sagittal plane angles at foot-strike and toe-off to error in step event detection by calculating the difference between the angle at the "ground truth" event obtained using the onset and offset of the vGRF method and the angle at five neighbouring time-points either side of it. These variables have been previously highlighted as key descriptors of running technique [28, 29]. The *Speed* dataset was used for this purpose because it included the widest range of speeds and it was, therefore, further processed to obtain full lower-limbs kinematics. Based on previous studies on the reliability of three-dimensional kinematics measured by motion capture in clinical applications [9] and in treadmill running [7], we considered an error in foot-strike or toe-off detection that implied a change of < 2° as acceptable, 2–5° as reasonable but requiring consideration and > 5° as unacceptable.

## Results

Cross-validation accuracy of FootNet was 98.96 ± 0.01%. The best performing set of parameters achieved a validation accuracy of 99.08% and was updated with the entire training dataset for final testing. FootNet achieved 99.23% accuracy in the classification of contact and non-contact frames on the test set. When comparing FootNet against the "gold standard" method in the detection of step events, there was a median bias of 0 (95LA = [−10, 7]) ms for foot-strike, 0 (95LA = [−10, 10]) ms for toe-off and 0 (95LA = [−15, 15]) ms for contact times (Fig 5). The RMSE was 5 ms for foot-strike, 6 ms for toe-off and 8 ms for contact time. Linear relations between the instant within the gait cycle where foot-strike and toe-off occurred, and errors in foot-strike and toe-off detection respectively were trivial. There was a small positive correlation between contact time length and error in contact time estimation (r = 0.1, p < 0.001, error = 0.032* average contact time between methods– 6.928, $r^2$ = 0.01).

The correlation analyses (Fig 6) revealed small negative linear associations between angle at foot-strike and error in foot-strike (r = -0.19, p < 0.001, error = -0.1447* angle at foot-strike + 0.486, $r^2$ = 0.04) and angle at foot-strike and error in contact time (r = 0.16, p < 0.001, error = 0.161 * angle at foot-strike + 0.166, $r^2$ = 0.02). Correlations between running speed and

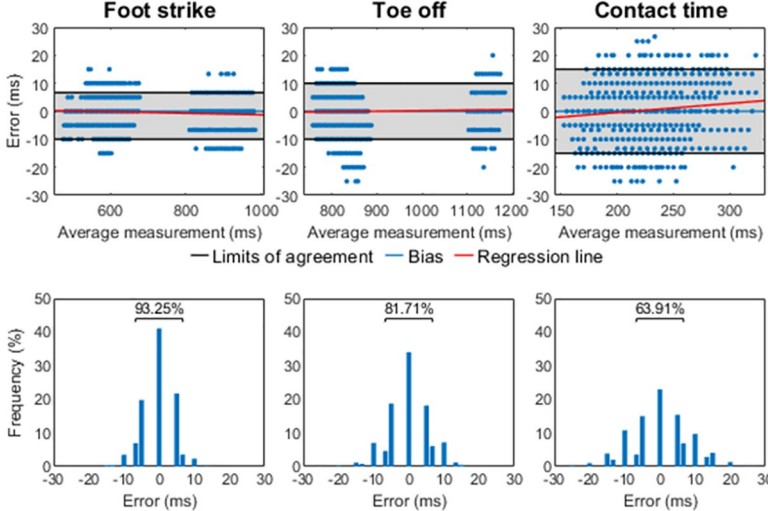

**Fig 5. Algorithm performance for foot-strike, toe-off and contact time.** Top row: Bland-Altman plots. The top and bottom border of the grey patch represent the 95% limits of agreement. Regression lines are included to aid visual interpretation. Correlations between instant within the gait cycle and error were trivial for foot-strike (r = -0.07, p < 0.001) and toe-off (r = 0.04, p < 0.001) hence, line coefficients and coefficient of determination are not reported. Average contact time length and error in contact time (r = 0.1, p < 0.001) line coefficients and coefficient of determination are reported in the Results section. Second row: error histograms. The frequency of errors contained within ±1 motion capture frame of the ground truth event is highlighted.

error in foot-strike, toe-off and contact time; and between incline and error in foot-strike, toe-off, and contact time were trivial.

The sensitivity analysis of sagittal plane angular kinematics to error in the detection of step events showed that angle errors were especially noticeable at toe-off (Fig 7 for an example at 4 m/s) and increased with speed (S1–S6 Figs). The knee angle was the most sensitive variable at foot-strike reaching unacceptable errors in ~15 ms after contact. Ankle angle was the most sensitive variable at toe-off with unacceptable errors when anticipating or delaying toe-off by ~10 ms. Anticipating toe-off by ~20 ms also led to unacceptable errors in hip and knee angle at speeds faster than 3.5 m/s.

## Discussion

The purpose of this study was to develop and evaluate FootNet, an algorithm for the detection of foot-strike and toe-off events in treadmill running based on lower limb kinematics. Our method is based on an RNN architecture with bidirectional LSTM units and has been developed and tested using five datasets collected in three independent laboratories under different experimental and running conditions. The proposed algorithm has achieved close agreement with the "gold standard" method for the detection of foot-strike and toe-off and outperforms previously published algorithms. The algorithm was also robust to different running speeds, foot-strike angle and incline as associations between algorithm performance and these factors were small and practically negligible. Errors in hip, knee and ankle sagittal plane angles due to the error in foot-strike and toe-off detection were mostly acceptable or reasonable when using FootNet.

FootNet uses a simple input (i.e. distal tibia anteroposterior velocity, ankle dorsi/plantar-flexion angle, and foot COM anteroposterior and vertical velocities) requiring minimal additional data processing for researchers. The use of segment kinematics makes our approach robust to problems with marker fixation faced by trajectory-based algorithms. The algorithm

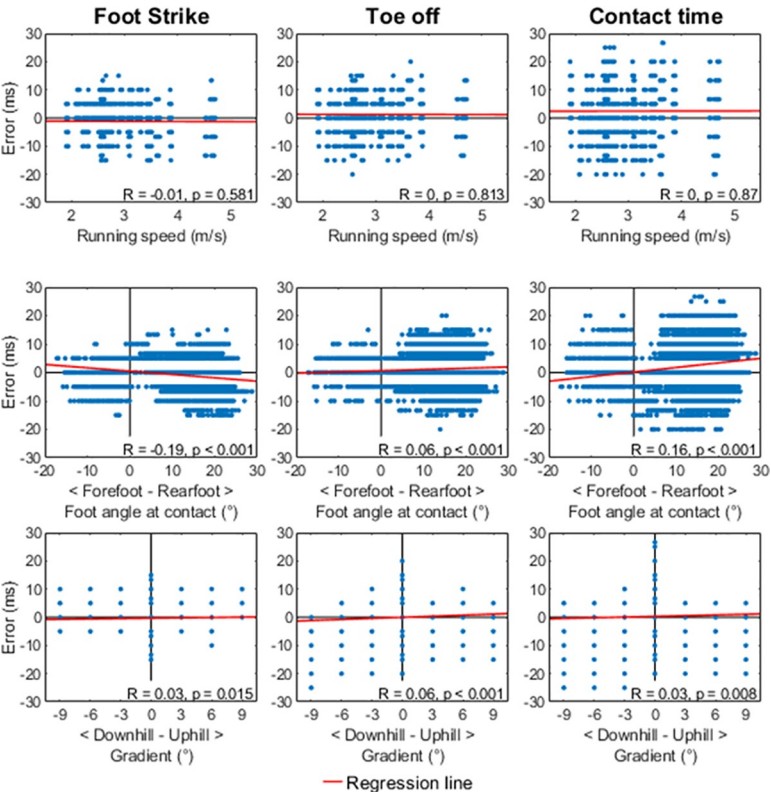

**Fig 6. Correlation analyses.** Scatter plots for the relation between running speed (top row), foot angle at contact (middle row) and incline (bottom row) and errors in the detection of foot-strike (left), toe-off (middle) and contact time (right) respectively. Note that only data on flat running surfaces was included for the running speed and foot angle at contact scatter plots and correlation analyses. Regression lines are included in every plot to aid visual interpretation, but line parameters and coefficient of determination are only reported in the Results section for nontrivial correlations.

has been trained using several foot tracking marker sets (Fig 1), allowing to place markers where fixation may not be as compromised as on the metatarsal heads. Since only the shank and foot segments are needed, it also overcomes problems related to complex marker sets requiring more segments. Additionally, FootNet has been trained and tested on data from runners of different levels, under different fatigue and shoe conditions and in three independent laboratories with different treadmills and motion capture systems. These sources of variability in training and testing data alongside 5-fold cross-validation and the overfitting prevention strategies implemented (i.e. dropout, early stopping) could explain the stable validation and testing accuracies. Validation and testing accuracies suggest that our model generalises well to unseen runners, overcoming issues encountered by previous algorithms [4] as reported by King et al. [3].

FootNet outperformed previously published algorithms for the detection of foot-strike and toe-off. The reported RMSE errors of 5 and 6 ms in the detection of step events were two to four times smaller than those reported by King et al. [3] when comparing different algorithms for the detection of foot-strike [4, 30] and toe-off [5]. This is also the case for contact times, with RMSE of 8 ms, which is two to three times smaller than previously published algorithms [4, 30]. Similarly, 95% of the errors when detecting foot-strike and toe-off were within ± 10 ms of the "gold standard" which is half the range previously reported by Osis et al. [10]. FootNet was robust to different running speeds and inclines with no linear associations between these

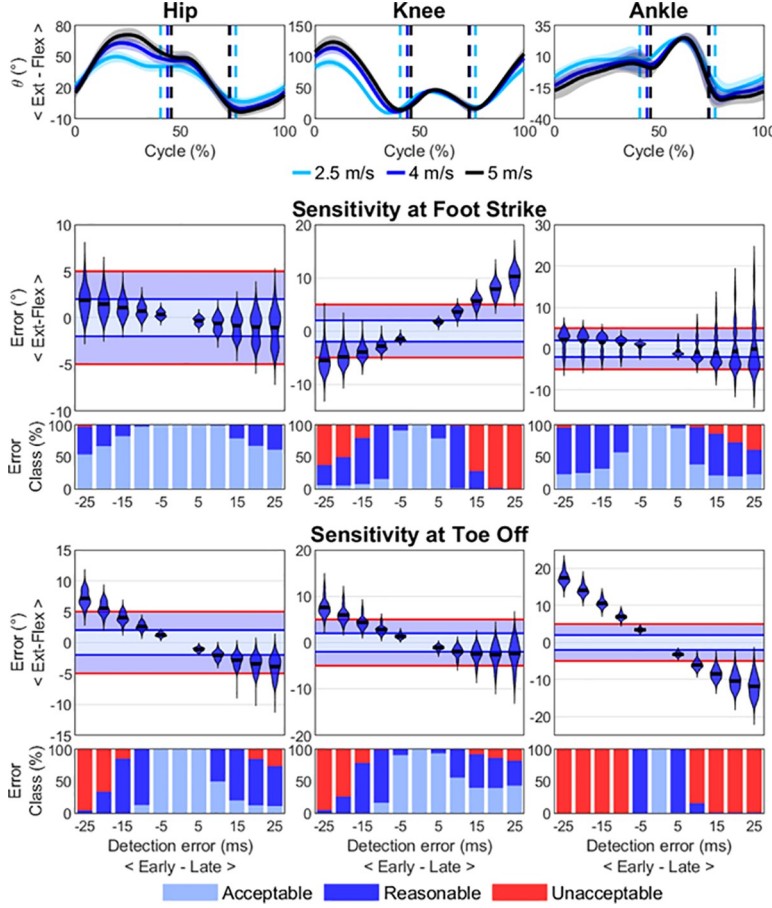

**Fig 7. Sensitivity analysis.** Top row: mean (± standard deviation) hip, knee and ankle sagittal plane angles at 2.5 (cyan), 4 (blue) and 5 (black) m/s during the gait cycle beginning from highest foot COM position to highest foot COM position. Average foot-strike and toe-off are indicated with vertical dashed lines to aid interpretation. More positive angles refer to dorsiflexion in the ankle plot. Using 4 m/s as example, the violin plots show the error (˚) distribution in hip, knee and ankle sagittal plane angles at foot-strike (2nd row) and toe-off (4th row) as a function of anticipated (negative time values on the x axis) or delayed (positive time values) step event detection, where the correct value of the variable is taken from the event time provided by the "gold standard" method (force plate). The horizontal black line within each violin represents the mean. Error classification in acceptable (light blue), reasonable (dark blue) and unacceptable (red) are displayed for foot-strike (third row) and toe-off (fifth row) respectively.

variables and error in foot-strike, toe-off or contact time estimation. Small linear associations between foot-strike angle and error in foot-strike and contact times were present. However, changes to foot-strike angle could only explain 4% and 2% of variance in foot-strike and contact time error respectively, and these associations may be deemed as practically negligible. FootNet appears to be applicable and equally effective at different running speeds, on different inclines and for runners with different foot-strike patterns without the need for foot-strike specific algorithms or speed corrections, overcoming some of the main limitations of kinematics-based step detection algorithms previously reported in the literature [3].

The sensitivity analysis of hip, knee and ankle sagittal plane angles to error in the detection of step events showed that angle errors increased with speed as expected and revealed that these errors were especially noticeable at toe-off. Knee angle was the most sensitive variable at foot-strike as previously found by Osis et al. [10], with changes >5˚ at ~15 ms after contact. The rapid extension of the hip, knee and ankle plantar flexion at the end of the stance phase made these variables especially sensitive to errors in toe-off detection. Ankle angle was the

most sensitive to error in toe-off detection with unacceptable errors occurring when anticipating or delaying toe-off detection by ~10 ms. We advise researchers investigating joint kinematics at foot-strike and toe-off using kinematic methods for the detection of step events to consider the angle errors we have reported to help interpret their results. Overall, the 95% limits of agreement between our algorithm and the "gold standard" method sat ~10 ms around the "ground truth" events, ensuring lower-limb joint angle errors remained acceptable (<2˚) or reasonable (2–5˚) for every variable but ankle angle at toe-off. Yet, 82% of toe-off events were detected within ~5 ms of the ground truth toe-off (Fig 5), ensuring most ankle angle errors remained reasonable.

## Limitations and perspectives

Using data from every dataset to maximise variability within the training data came at the cost of not having a testing set collected under completely different laboratory conditions. The Speed, Footwear and Prolonged datasets were collected in the same laboratory and put together, provided the largest amount of data. Hence, leaving one of these datasets out for testing would be ineffective, and leaving the three of them out could hinder training capacity. The Foot-strikes dataset provided the largest number of non-rearfoot strikers, which was limited in the other datasets and the Inclines dataset provided different treadmill slopes, which were only present within this dataset. We considered mixing the datasets and leaving 30% participants from each set out for testing the best compromise between variability in training and training-testing data independence.

Force-instrumented treadmills are usually stiffer than non-instrumented ones where our algorithm is intended to be used. Different surface mechanical properties can affect running kinematics [31] and, unfortunately, there are currently no recommended standards for treadmills used in research. Although algorithm performance may be affected on different treadmills, an advantage of using a deep learning approach is that the algorithm can be further optimised with new datasets. For instance, high-speed cameras could be used to assess and improve the validity of our algorithm on non-instrumented treadmills with different mechanical properties. A transfer learning framework [32] could also be developed whereby a researcher can manually label a relatively small number of cycles and use those to re-optimise FootNet's parameters for a specific dataset collected under the same conditions. For these reasons and in the interest of further improving the tool for the biomechanics research community, FootNet and its source code is available for download on GitHub (https://github.com/adrianrivadulla/FootNet) and contributions are encouraged.

## Conclusions

We have developed and evaluated FootNet, a new and opensource algorithm that improves the detection of foot-strike and toe-off events based on kinematics for treadmill running. This algorithm requires a simple input and outperforms previously published algorithms for the detection of step events. FootNet is also robust to different running speeds, foot-strike patterns and inclines. Errors in kinematic variables as a result of errors in foot-strike and toe-off detection using FootNet were mostly classified as acceptable. The algorithm is publicly available, and its use is recommended when studying treadmill running biomechanics in the absence of force plates.

## Supporting information

**S1 Fig. Sensitivity analysis at 2.5 m/s.** Top row: mean (± standard deviation) hip, knee and ankle sagittal plane angles during the gait cycle beginning from highest foot COM position to

highest foot COM position. Average foot strike and toe off are indicated with vertical dashed lines to aid interpretation. More positive angles refer to dorsiflexion in the ankle plot. The violin plots show the error (˚) distribution in hip, knee and ankle sagittal plane angles at foot strike (2nd row) and toe off (4th row) as a function of anticipated (negative time values on the x axis) or delayed (positive time values) step event detection, where the correct value of the variable is taken from the event time provided by the "gold standard" method (force plate). The horizontal black line within each violin represents the mean. Error classification in acceptable (light blue), reasonable (dark blue) and unacceptable (red) are displayed for foot strike (third row) and toe off (fifth row) respectively.
(DOCX)

**S2 Fig. Sensitivity analysis at 3 m/s.** Top row: mean (± standard deviation) hip, knee and ankle sagittal plane angles during the gait cycle beginning from highest foot COM position to highest foot COM position. Average foot strike and toe off are indicated with vertical dashed lines to aid interpretation. More positive angles refer to dorsiflexion in the ankle plot. The violin plots show the error (˚) distribution in hip, knee and ankle sagittal plane angles at foot strike (2nd row) and toe off (4th row) as a function of anticipated (negative time values on the x axis) or delayed (positive time values) step event detection, where the correct value of the variable is taken from the event time provided by the "gold standard" method (force plate). The horizontal black line within each violin represents the mean. Error classification in acceptable (light blue), reasonable (dark blue) and unacceptable (red) are displayed for foot strike (third row) and toe off (fifth row) respectively.
(DOCX)

**S3 Fig. Sensitivity analysis at 3.5 m/s.** Top row: mean (± standard deviation) hip, knee and ankle sagittal plane angles during the gait cycle beginning from highest foot COM position to highest foot COM position. Average foot strike and toe off are indicated with vertical dashed lines to aid interpretation. More positive angles refer to dorsiflexion in the ankle plot. The violin plots show the error (˚) distribution in hip, knee and ankle sagittal plane angles at foot strike (2nd row) and toe off (4th row) as a function of anticipated (negative time values on the x axis) or delayed (positive time values) step event detection, where the correct value of the variable is taken from the event time provided by the "gold standard" method (force plate). The horizontal black line within each violin represents the mean. Error classification in acceptable (light blue), reasonable (dark blue) and unacceptable (red) are displayed for foot strike (third row) and toe off (fifth row) respectively.
(DOCX)

**S4 Fig. Sensitivity analysis at 4 m/s.** Top row: mean (± standard deviation) hip, knee and ankle sagittal plane angles during the gait cycle beginning from highest foot COM position to highest foot COM position. Average foot strike and toe off are indicated with vertical dashed lines to aid interpretation. More positive angles refer to dorsiflexion in the ankle plot. The violin plots show the error (˚) distribution in hip, knee and ankle sagittal plane angles at foot strike (2nd row) and toe off (4th row) as a function of anticipated (negative time values on the x axis) or delayed (positive time values) step event detection, where the correct value of the variable is taken from the event time provided by the "gold standard" method (force plate). The horizontal black line within each violin represents the mean. Error classification in acceptable (light blue), reasonable (dark blue) and unacceptable (red) are displayed for foot strike (third row) and toe off (fifth row) respectively.
(DOCX)

**S5 Fig. Sensitivity analysis at 4.5 m/s.** Top row: mean (± standard deviation) hip, knee and ankle sagittal plane angles during the gait cycle beginning from highest foot COM position to highest foot COM position. Average foot strike and toe off are indicated with vertical dashed lines to aid interpretation. More positive angles refer to dorsiflexion in the ankle plot. The violin plots show the error (˚) distribution in hip, knee and ankle sagittal plane angles at foot strike (2nd row) and toe off (4th row) as a function of anticipated (negative time values on the x axis) or delayed (positive time values) step event detection, where the correct value of the variable is taken from the event time provided by the "gold standard" method (force plate). The horizontal black line within each violin represents the mean. Error classification in acceptable (light blue), reasonable (dark blue) and unacceptable (red) are displayed for foot strike (third row) and toe off (fifth row) respectively.
(DOCX)

**S6 Fig. Sensitivity analysis at 5 m/s.** Top row: mean (± standard deviation) hip, knee and ankle sagittal plane angles during the gait cycle beginning from highest foot COM position to highest foot COM position. Average foot strike and toe off are indicated with vertical dashed lines to aid interpretation. More positive angles refer to dorsiflexion in the ankle plot. The violin plots show the error (˚) distribution in hip, knee and ankle sagittal plane angles at foot strike (2nd row) and toe off (4th row) as a function of anticipated (negative time values on the x axis) or delayed (positive time values) step event detection, where the correct value of the variable is taken from the event time provided by the "gold standard" method (force plate). The horizontal black line within each violin represents the mean. Error classification in acceptable (light blue), reasonable (dark blue) and unacceptable (red) are displayed for foot strike (third row) and toe off (fifth row) respectively.
(DOCX)

## Acknowledgments

The authors would like to thank Dr. R. K. Fukuchi, Dr. C. A. Fukuchi, Dr. M. Duarte, Ms. E. S. Matijevich, Ms. L. M. Branscombe, Dr. L. R. Scott and Dr. K. E. Zelik for making their datasets open access. We would also like to thank Dr. Laurie Needham for his continuous support and advice on algorithm development and John Kuzmeski and Alexis Alecia for their contribution to data collection and data processing.

## Author Contributions

**Conceptualization:** Adrian Rivadulla, Xi Chen, Gillian Weir, Dario Cazzola, Grant Trewartha, Joseph Hamill, Ezio Preatoni.

**Data curation:** Adrian Rivadulla, Gillian Weir.

**Formal analysis:** Adrian Rivadulla, Xi Chen, Ezio Preatoni.

**Investigation:** Adrian Rivadulla, Xi Chen, Gillian Weir, Ezio Preatoni.

**Methodology:** Adrian Rivadulla, Xi Chen, Ezio Preatoni.

**Project administration:** Ezio Preatoni.

**Resources:** Adrian Rivadulla, Gillian Weir, Joseph Hamill.

**Software:** Adrian Rivadulla, Xi Chen.

**Supervision:** Xi Chen, Gillian Weir, Dario Cazzola, Grant Trewartha, Joseph Hamill, Ezio Preatoni.

**Validation:** Adrian Rivadulla, Xi Chen, Gillian Weir, Dario Cazzola, Grant Trewartha, Joseph Hamill, Ezio Preatoni.

**Visualization:** Adrian Rivadulla.

**Writing – original draft:** Adrian Rivadulla, Xi Chen, Gillian Weir, Dario Cazzola, Grant Trewartha, Joseph Hamill, Ezio Preatoni.

**Writing – review & editing:** Xi Chen, Gillian Weir, Dario Cazzola, Grant Trewartha, Joseph Hamill, Ezio Preatoni.

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
