## [Decision Letter · Decision Letter 0]

22 Apr 2021

PONE-D-21-06501

Development and validation of FootNet; a new kinematic algorithm to improve foot-strike and toe-off detection in treadmill running

PLOS ONE

Dear Dr. Rivadulla,

Thank you for submitting your manuscript to PLOS ONE. After careful consideration, we feel that it has merit but does not fully meet PLOS ONE’s publication criteria as it currently stands. Therefore, we invite you to submit a revised version of the manuscript that addresses the points raised during the review process.

We look forward to receiving your revised manuscript.

Kind regards,

Laurent Mourot

Academic Editor

PLOS ONE

Journal Requirements:

2. Thank you for including your ethics statement:  "Ethical approval was granted by the home institutions where the data were originally collected".  

a.) Please amend your current ethics statement to include the full name of the ethics committee/institutional review board(s) that approved your specific study.

b.) Please provide additional details regarding participant consent. In the ethics statement in the Methods and online submission information, please ensure that you have specified (1) whether consent was informed and (2) what type you obtained (for instance, written or verbal, and if verbal, how it was documented and witnessed). If your study included minors, state whether you obtained consent from parents or guardians. If the need for consent was waived by the ethics committee, please include this information.

3. We noted in your submission details that a portion of your manuscript may have been presented or published elsewhere. The raw data used in this study have previously been used for other studies. Two of the datasets we have used are open access datasets and the other three were provided by the co-authors. Please clarify whether this publication was peer-reviewed and formally published. If this work was previously peer-reviewed and published, in the cover letter please provide the reason that this work does not constitute dual publication and should be included in the current manuscript.

5. Please ensure that you refer to Figure 7 in your text as, if accepted, production will need this reference to link the reader to the figure.

6. Please include your tables as part of your main manuscript and remove the individual files. Please note that supplementary tables (should remain/ be uploaded) as separate "supporting information" files

Reviewers' comments:

Reviewer's Responses to Questions

**Comments to the Author**

1. Is the manuscript technically sound, and do the data support the conclusions?

Reviewer #1: Yes

Reviewer #2: Yes

Reviewer #3: Yes

2. Has the statistical analysis been performed appropriately and rigorously? 

Reviewer #1: Yes

Reviewer #2: Yes

Reviewer #3: Yes

3. Have the authors made all data underlying the findings in their manuscript fully available?

Reviewer #1: Yes

Reviewer #2: No

Reviewer #3: No

4. Is the manuscript presented in an intelligible fashion and written in standard English?

Reviewer #1: Yes

Reviewer #2: Yes

Reviewer #3: Yes

5. Review Comments to the Author

Reviewer #1: I loved reading this paper. It is well-organised, with a nice introduction that sets the scene and with a very thought-out discussion that can put results in context. Pretty everything that I was looking for in this manuscript was right there where it was supposed to be. The authors should be commended for the work they have done. Additional contribution with code and dataset (and perhaps model weights or checkpoint) might constitute a real additional value.

Strength points are constituted by:

1) a large (and open) dataset and an open git project

2) a cross-fold validation analysis

3) state of the art algorithm such as LSTM

4) excellent statistical approach to discuss the results and put-in-context (with magnitude based inferences)

My overall impression about the manuscript is really positive, and I only have few general and specific comments.

GENERAL COMMENTS

Interestingly you conducted the regression analysis with an output layer which is of the same length of the input. This looks like an encoding problem, in which you look for a reduction in the dimension of the input to provide a light representation of the input signals. Interestingly, many other authors look for a solution where they select a sequence length and a stride length for a moving window, and they train a classifier to detect contact/non-contact phases in just few instants at the end of the window. It’s interesting to notice that if you use a stride length which is too short you might end up with too many samples and a trivial dataset. Conversely, if you do not select these windows appropriately you might miss a system dynamic (if steps are way faster than windowing stride). On the other hand, the encoding problem you are trying to solve requires a considerable amount of output neurons.

I think that the description of the dataset and related processes could be expanded a bit. Particularly, I think that contact phases were longer than aerial phases, hence an unbalanced dataset, is this correct (maybe I’m wrong)? How did you deal with this issue, if pertinent? Did you apply any dataset augmentation technique? I’m also impressed that I cannot find any reference to the data normalisation/standardisation technique you adopted, if any. Please notice that I’m talking about the input signals here.

I acknowledge the fact that no strict rules about the development of an architecture exists. But I’m wondering if you could be more specific on how you ended up with your network architecture. Did you try to find a trade-off between accuracy and computational time, or you just took this architecture as it was already working. Notice that joining more NN is always possible. Especially if you take a NN which has been already trained, so you can cut out the last layer and grab its output features.

Computational and inference times: did you have any chance to connect your training/validation process with a Wandb account for instance, where users might be able to see training progresses and training accuracy/validation in real time? I’m also wondering how long would an inference take, this is just because thinking about real time applications. What if we were able to connect the model to a motion capture system for instance, and to provide inference in quasi-real-time/real-time?

SPECIFIC

L54: I would argue that other measurement systems might constitute the gold-standard. E.g. those systems working with light-obstruction principles? E.g. Optogait analysis?

L80: would it be possible, in your opinion, to put this information in context and evaluate a % error based on typical gait patterns? This might constitute additional relevant information for the reader.

L103: This is just a comment. In my opinion LSTM are a great choice for time-series analysis, but I would consider that also CNN are used to solve regression/classification problems in time-series.

L201: maybe I’m missing something, but in light with my generic comment, could you please expand a bit on the why you selected 200?

Bland-Altman plots: it looks that there is a resolution issue (points scattered along lines) in these plots. And this is usually due to sampling frequencies. Any comment on that?

Reviewer #2: Comments:

Thank you for the opportunity to review this well written and intriguing manuscript. The author examined novel kinematics and deep learning-based algorithm for the detection of step events in treadmill running. Deep learning-based algorithm is sound. Overall, I have minor comments for the authors.

・It is true that conventional marker-based algorithms may be affected by the deformable area. However, validity has been confirmed in previous studies. For example, Smith et al. (A comparison of kinematic algorithms to estimate gait events during overground running) algorithm has much less error than force plate. In order for the reader to use Foot net, it is necessary to state Introduction and Discussion that it is a more appropriate method compared to the methods of other previous studies.

Smith et al. (2015)

・Line 178: shankwas. Please divide.

・All formulas are hard to read.(e.g. line 212-214, line 216, line 218-219...). I recommend use Word's equation editor.

・Line 479: I can't download Github. “This is not the web page” is displayed

Reviewer #3: The Authors present a nice technique to accurately identify the stance phase of running using machine learning. This technique seems to be robust across a range of experimentally relevant conditions including speed, incline, shoe-type, foot strike, and heel/toe strike. The study is well justified and a clear improvement on existing techniques. My area of expertise is in lower extremity biomechanics not machine learning, so I will defer to other Reviewers for more technical critiques. However, the end result seems quite robust and it is understandable that a separate validation cohort wasn’t leveraged given COVID restrictions. I don’t have any major concerns with this study and think it will make a nice addition to the literature.

Minor concerns.

-Include regulatory info on the Speed, Footwear, and Fatigue data sets. Was informed consent acquired? Do these data appear elsewhere?

-it is understable that joint kinematics are impacted by small errors in foot contact and toe off timing. Based on your findings, it might be helpful to provide guidance to the number of strides researchers should plan on analyzing so these errors average out to zero. This seems very plausible given that the average timing errors were 0ms – which is very impressive. Since FootNet is meant for treadmill running, it seems like researchers are going to collect enough strides to adequately resolve these kinematic errors. Some firm guidance would strengthen the use of FootNet.

-From a tissue-loading perspective, getting foot contact timing is probably more important than toe-off timing. For example, during weight acceptance tissues are being rapidly loaded – which may be associated with injury risk. But during toe-off, the body is rapidly leaving the ground and tissue loads are more gradually decreasing. This type of perspective might help readers appreciate the practical implications of errors at contact and toe-off.

-the github url throws a 404 error. Please get this posted.

6. PLOS authors have the option to publish the peer review history of their article (what does this mean?). If published, this will include your full peer review and any attached files.

Reviewer #1: **Yes: **Andrea Zignoli

Reviewer #2: No

Reviewer #3: **Yes: **Josh R Baxter

---

## [Author Response · Author response to Decision Letter 0]

21 May 2021

Response to reviewers for the attention of the Editor and reviewers of the manuscript titled:

“Development and validation of FootNet; a new kinematic algorithm to improve foot-strike and toe-off detection in treadmill running”

The authors would like to thank the Editor and reviewers for their work and kind words about our study. We believe the suggestions provided have contributed to improve the quality and clarity of our manuscript. In this letter, we address each of the comments, outlining the changes made to the previously submitted version of our manuscript and adding an explanation when deemed needed. All the changes to the manuscript have been highlighted in yellow in the “with Tracked Changes” copy.

Manuscript Decision

Journal Requirements:

 Response. Done. 

2. Thank you for including your ethics statement: "Ethical approval was granted by the home institutions where the data were originally collected". 

a.) Please amend your current ethics statement to include the full name of the ethics committee/institutional review board(s) that approved your specific study.

 Response. Done

b.) Please provide additional details regarding participant consent. In the ethics statement in the Methods and online submission information, please ensure that you have specified (1) whether consent was informed and (2) what type you obtained (for instance, written or verbal, and if verbal, how it was documented and witnessed). If your study included minors, state whether you obtained consent from parents or guardians. If the need for consent was waived by the ethics committee, please include this information.

 Response. Done.

 Response. Done.

3. We noted in your submission details that a portion of your manuscript may have been presented or published elsewhere. The raw data used in this study have previously been used for other studies. Two of the datasets we have used are open access datasets and the other three were provided by the co-authors. Please clarify whether this publication was peer-reviewed and formally published. If this work was previously peer-reviewed and published, in the cover letter please provide the reason that this work does not constitute dual publication and should be included in the current manuscript.

 Response. The two open access datasets named “Foot Strikes” and “Inclines” in the manuscript were previously used for other research projects published in peer-reviewed scientific journals by the original authors as referenced in the manuscript. Our present submission does not constitute dual publication because the purpose of our research is completely different to the purpose for which the datasets were collected and additionally, we reprocessed the raw data files provided for our specific study. We have no formal (nor informal for that matter) relation with the original authors of these two datasets, we are just benefitting from their decision to embrace the open science philosophy and to make their data accessible to everyone, reason for which we show our gratitude in the Acknowledgements.

 Response. Our dataset has been uploaded to the University of Bath repository and will be made publicly available upon full acceptance of the paper. The reserved DOI for our dataset is: https://doi.org/10.15125/BATH-00965. Note that as of right now, this doi will not work until full access is activated by our librarians. We do have a temporary link that Editor and reviewers can use to access the data: 

https://files.bath.ac.uk/index.php/s/XYarEqCye8H7GfR. 

Password: xYX3txZdGEEqvVFj

This link will expire on 18th June but please get in contact to get a new one if needed. We apologise for this inconvenience but apparently it is the standard procedure to give access to unpublished repository content at the University of Bath for security reasons.

5. Please ensure that you refer to Figure 7 in your text as, if accepted, production will need this reference to link the reader to the figure.

 Response. Done

6. Please include your tables as part of your main manuscript and remove the individual files. Please note that supplementary tables (should remain/ be uploaded) as separate "supporting information" files

 Response. Done.

 Response. I am not entirely sure if I have done it correctly.

Reviewers' comments:

Reviewer's Responses to Questions

Comments to the Author

1. Is the manuscript technically sound, and do the data support the conclusions?

Reviewer #1: Yes

Reviewer #2: Yes

Reviewer #3: Yes

2. Has the statistical analysis been performed appropriately and rigorously? 

Reviewer #1: Yes

Reviewer #2: Yes

Reviewer #3: Yes

3. Have the authors made all data underlying the findings in their manuscript fully available?

Reviewer #1: Yes

Reviewer #2: No

Reviewer #3: No

4. Is the manuscript presented in an intelligible fashion and written in standard English?

Reviewer #1: Yes

Reviewer #2: Yes

Reviewer #3: Yes

5. Review Comments to the Author

Reviewer #1: I loved reading this paper. It is well-organised, with a nice introduction that sets the scene and with a very thought-out discussion that can put results in context. Pretty everything that I was looking for in this manuscript was right there where it was supposed to be. The authors should be commended for the work they have done. Additional contribution with code and dataset (and perhaps model weights or checkpoint) might constitute a real additional value.

Strength points are constituted by:

1) a large (and open) dataset and an open git project

2) a cross-fold validation analysis

3) state of the art algorithm such as LSTM

4) excellent statistical approach to discuss the results and put-in-context (with magnitude based inferences)

My overall impression about the manuscript is really positive, and I only have few general and specific comments.

GENERAL COMMENTS

Interestingly you conducted the regression analysis with an output layer which is of the same length of the input. This looks like an encoding problem, in which you look for a reduction in the dimension of the input to provide a light representation of the input signals. Interestingly, many other authors look for a solution where they select a sequence length and a stride length for a moving window, and they train a classifier to detect contact/non-contact phases in just few instants at the end of the window. It’s interesting to notice that if you use a stride length which is too short you might end up with too many samples and a trivial dataset. Conversely, if you do not select these windows appropriately you might miss a system dynamic (if steps are way faster than windowing stride). On the other hand, the encoding problem you are trying to solve requires a considerable amount of output neurons.

Response. Due to the cyclic nature of running and under normal circumstances, our “windowing” (mid-swing to mid-swing) of the signal should always include a non-phase followed by a contact phase and another non-contact phase. Mid-swing detection was visually assessed and in fact did not provide any issue.

I think that the description of the dataset and related processes could be expanded a bit. Particularly, I think that contact phases were longer than aerial phases, hence an unbalanced dataset, is this correct (maybe I’m wrong)? How did you deal with this issue, if pertinent? Did you apply any dataset augmentation technique? I’m also impressed that I cannot find any reference to the data normalisation/standardisation technique you adopted, if any. Please notice that I’m talking about the input signals here.

 Response. This is a very good point as it is true that the aerial phase is typically longer than the contact phase in running hence, the non-contact class can indeed be over-represented. This may partly explain the high classification accuracy (contact or non-contact frames) which is what the neural network is trained for). Certainly, some data augmentation e.g. truncating the stride cycle so the aerial phase was as long as the contact phase could be implemented, potentially leading to a “less inflated” accuracy. However, the purpose of the complete algorithm truly comes down to finding the first and last frames classified as contact. Thus, we decided not to emphasise the accuracy achieved, even though still relevant to be reported as this was a classification problem, and we focused our results and discussion on the detection of foot-strike and toe-off. Besides, we included signals captured at different sampling frequencies and runners with different running styles ensuring that, although contact phase may always be shorter than the aerial phase, the proportion of contact and non-contact frames and the start and end of the contact phase should vary within the training examples. Input signals were standardised to z-scores as indicated in lines 254-257.

I acknowledge the fact that no strict rules about the development of an architecture exists. But I’m wondering if you could be more specific on how you ended up with your network architecture. Did you try to find a trade-off between accuracy and computational time, or you just took this architecture as it was already working. Notice that joining more NN is always possible. Especially if you take a NN which has been already trained, so you can cut out the last layer and grab its output features.

Response. Model architecture was developed from scratch for the purpose of our study. We prioritised accuracy and we did not consider computational time (as long as it was reasonable) as the primary use of this algorithm would be in post-data collection processing rather than live or quasi-live data processing. 

Computational and inference times: did you have any chance to connect your training/validation process with a Wandb account for instance, where users might be able to see training progresses and training accuracy/validation in real time? I’m also wondering how long would an inference take, this is just because thinking about real time applications. What if we were able to connect the model to a motion capture system for instance, and to provide inference in quasi-real-time/real-time?

Response. Training progress and training/validation accuracy can be tracked in real time as the fit method in Tensorflow allows the user to control the verbosity of the process. By default, epoch count, a progress bar, time to complete each epoch and training and validation accuracy get displayed on the command line as training proceeds. Real time implementation is certainly something to consider in the future but that may be infeasible as things stand right now. The input signals require filtering, rigid body modelling, signal standardisation, padding and all the operation involved in the neural network. These processes take time and that is why we believe real time implementation of this algorithm was out of scope and would require a considerable amount of adaptations and re-evaluation. For instance, a file of 27 strides (see example file in repository) can take several seconds to be fully processed in an average CPU (6-core i7 processor, 16GB RAM). As indicated in the above question, we favoured accuracy over speed in this case as we were interested in developing a tool for more comprehensive data processing and analysis that does not require real time processing.

SPECIFIC

L54: I would argue that other measurement systems might constitute the gold-standard. E.g. those systems working with light-obstruction principles? E.g. Optogait analysis?

Response. Although Optogait is a valid method for the estimation of gait spatiotemporal parameters, the literature seems to suggest that force data is the gold standard for contact events. Indeed, all the previous studies on step detection algorithm development referenced in our manuscript use force data as their gold standard.

L80: would it be possible, in your opinion, to put this information in context and evaluate a % error based on typical gait patterns? This might constitute additional relevant information for the reader.

Response. We think this is an interesting point, however the authors of that study did not provide the entire knee flex-ext angle curve so it is not possible to express that error as a percentage of the knee range of motion. We did check it out of curiosity after your suggestion, using our data at 2.5 m/s, which is the closest one we have to their reported average speed (2.65 ± 0.22 m/s) and an error of 7 degrees can mean up to 20% of the range of motion in the contact phase or about 8% in the full stride range of motion. 

L103: This is just a comment. In my opinion LSTM are a great choice for time-series analysis, but I would consider that also CNN are used to solve regression/classification problems in time-series.

Response. We fully agree, there are numerous examples in the literature where CNNs outperform LSTMs for time-series analysis. The advantage we saw in using LSTM is that it is flexible to take input signals of variable length in implementation phase, being able to accommodate stride cycles of different length as they naturally occur without the need of a time registration.

L201: maybe I’m missing something, but in light with my generic comment, could you please expand a bit on the why you selected 200?

Response. Zero-padding was selected to allow the formation of training data batches that can be passed on to Tensorflow’s fit method as tensors of a fixed shape, allowing for faster computation times. 200 datapoints was selected because it ensures that we have at least the equivalent to one second long worth of data at the motion capture sampling frequencies we had in the dataset (150 Hz and 200 Hz). Considering the running speeds present in the dataset, a stride should hardly ever reach that time length thus, 200 seemed like a sensible choice to accommodate every possible stride length. Note that zero-padding is not needed in implementation.

Bland-Altman plots: it looks that there is a resolution issue (points scattered along lines) in these plots. And this is usually due to sampling frequencies. Any comment on that?

Response. Scattering of the points is indeed related to the motion capture sampling frequency. This is an interesting point by the reviewer with no single correct answer as it depends on what a researcher is interested in. We believe that it is up to the researcher to decide how granular their step detection and contact time measurements need to be for their specific research question. If subtle changes in hip, knee or especially the ankle as per our sensitivity analysis or contact times are the subject of study, perhaps choosing a motion capture system with a higher sampling frequency (and having the computing resources to be able to capture and process great amounts of data) or a different system that can sample a higher frequencies e.g. Optogait should be considered.

Reviewer #2: Comments:

Thank you for the opportunity to review this well written and intriguing manuscript. The author examined novel kinematics and deep learning-based algorithm for the detection of step events in treadmill running. Deep learning-based algorithm is sound. Overall, I have minor comments for the authors.

・It is true that conventional marker-based algorithms may be affected by the deformable area. However, validity has been confirmed in previous studies. For example, Smith et al. (A comparison of kinematic algorithms to estimate gait events during overground running) algorithm has much less error than force plate. In order for the reader to use Foot net, it is necessary to state Introduction and Discussion that it is a more appropriate method compared to the methods of other previous studies.

Smith et al. (2015)

 Response. We acknowledge that this could be a bit unclear in the version we have previously submitted. Our point is not that markers being on deformable areas can affect the performance of the algorithm because as you indicate, there are multiple algorithms validated with markers on such locations. We were trying to emphasise that the fixation of markers on highly deformable areas can be compromised especially in longer trials such as those collected on treadmills, and loose or fully detached markers do create problems. This is something we have seen in our experience studying running biomechanics and it was one of the main motivations to complete this study. We have tried to make this more explicit in line 68.

・Line 178: shankwas. Please divide.

 Response. Done.

・All formulas are hard to read.(e.g. line 212-214, line 216, line 218-219...). I recommend use Word's equation editor.

 Response. We used the Word’s equation editor to write all our equations. Perhaps this is something the manuscript formatting team from Plos One can help us with if they still do not look quite right?

・Line 479: I can't download Github. “This is not the web page” is displayed

Response. We apologise for this and it is certainly something we will reconsider in future occasions as there were also people who read the pre-print in Biorxiv who also contacted us with the same issue. Our thinking here was to wait for the paper to be accepted to make the repository public to make sure our study has passed all the standard scrutiny of the scientific process. All the code stored on Github can be access through the temporary link to the data repository: 

https://files.bath.ac.uk/index.php/s/XYarEqCye8H7GfR. 

Password: xYX3txZdGEEqvVFj

This link will expire on 18th June but please get in contact to get a new one if needed. We apologise for this inconvenience but apparently it is the standard procedure to give access to unpublished repository content at the University of Bath for security reasons.

Reviewer #3: The Authors present a nice technique to accurately identify the stance phase of running using machine learning. This technique seems to be robust across a range of experimentally relevant conditions including speed, incline, shoe-type, foot strike, and heel/toe strike. The study is well justified and a clear improvement on existing techniques. My area of expertise is in lower extremity biomechanics not machine learning, so I will defer to other Reviewers for more technical critiques. However, the end result seems quite robust and it is understandable that a separate validation cohort wasn’t leveraged given COVID restrictions. I don’t have any major concerns with this study and think it will make a nice addition to the literature.

Minor concerns.

-Include regulatory info on the Speed, Footwear, and Fatigue data sets. Was informed consent acquired? Do these data appear elsewhere?

Response. Details on ethical approval and participants’ consent have been included for each dataset. The data from the Speed, Footwear and Fatigue datasets were originally collected for other studies, two of which are currently under review elsewhere. However, the purpose and analytical approach taken in those are completely independent to the present study.

-it is understable that joint kinematics are impacted by small errors in foot contact and toe off timing. Based on your findings, it might be helpful to provide guidance to the number of strides researchers should plan on analyzing so these errors average out to zero. This seems very plausible given that the average timing errors were 0ms – which is very impressive. Since FootNet is meant for treadmill running, it seems like researchers are going to collect enough strides to adequately resolve these kinematic errors. Some firm guidance would strengthen the use of FootNet.

 Response. This is an interesting point but note that the median bias and the limits of agreement were calculated including every single stride from every participant in the testing set thus, the errors within a trial for one participant may not necessarily average out or approach 0. As an example, if we had collected 101 strides from 10 participants (nine have 10 strides, one has 11), 50 strides were -50 ms off, 50 strides were 50 ms off and one stride was 0 ms off, the median will still be 0. Since this error may be associated with experimental errors beyond the control of the algorithm, we believe that this point is certainly something to consider in future developments but beyond of the scope of the current study.

-From a tissue-loading perspective, getting foot contact timing is probably more important than toe-off timing. For example, during weight acceptance tissues are being rapidly loaded – which may be associated with injury risk. But during toe-off, the body is rapidly leaving the ground and tissue loads are more gradually decreasing. This type of perspective might help readers appreciate the practical implications of errors at contact and toe-off.

 Response. We appreciate the suggestion as greater errors in toe-off compared to foot-strike was a trend we noticed throughout model development and most model hyperparameter tweaking was to try to correct for larger errors in toe off compared to foot strike. Eventually, we reached very similar results in terms of limits of agreement (foot-strike: 95LA = [-10, 7], RMSE = 5 ms; and toe off: 95:A = [-10, 10], RMSE = 6 ms). Although foot-strike detection appears marginally better, if those results are translated to frames at for instance 200 Hz which is ultimately what a researcher is going to be working with, limits of agreement for both foot-strike and toe-off approach two frames and RMSE for both approach one frame. Therefore, we considered this marginally better result for foot-strike detection practically irrelevant.

-the github url throws a 404 error. Please get this posted.

Response. We apologise for this and it is certainly something we will reconsider in future occasions as there were also people who read the pre-print in Biorxiv who also contacted us with the same issue. Our thinking here was to wait for the paper to be accepted to make the repository public to make sure our study has passed all the standard scrutiny of the scientific process. All the code stored on Github can be access through the temporary link to the data repository: 

https://files.bath.ac.uk/index.php/s/XYarEqCye8H7GfR. 

Password: xYX3txZdGEEqvVFj

This link will expire on 18th June but please get in contact to get a new one if needed. We apologise for this inconvenience but apparently it is the standard procedure to give access to unpublished repository content at the University of Bath for security reasons.

6. PLOS authors have the option to publish the peer review history of their article (what does this mean?). If published, this will include your full peer review and any attached files.

Do you want your identity to be public for this peer review? For information about this choice, including consent withdrawal, please see our Privacy Policy.

Reviewer #1: Yes: Andrea Zignoli

Reviewer #2: No

Reviewer #3: Yes: Josh R Baxter

---

## [Decision Letter · Decision Letter 1]

5 Jul 2021

Development and validation of FootNet; a new kinematic algorithm to improve foot-strike and toe-off detection in treadmill running

PONE-D-21-06501R1

Dear Dr. Rivadulla,

We’re pleased to inform you that your manuscript has been judged scientifically suitable for publication and will be formally accepted for publication once it meets all outstanding technical requirements.

Kind regards,

Laurent Mourot

Academic Editor

PLOS ONE

Additional Editor Comments (optional):

Reviewers' comments:

Reviewer's Responses to Questions

**Comments to the Author**

1. If the authors have adequately addressed your comments raised in a previous round of review and you feel that this manuscript is now acceptable for publication, you may indicate that here to bypass the “Comments to the Author” section, enter your conflict of interest statement in the “Confidential to Editor” section, and submit your "Accept" recommendation.

Reviewer #1: All comments have been addressed

Reviewer #2: All comments have been addressed

2. Is the manuscript technically sound, and do the data support the conclusions?

Reviewer #1: Yes

Reviewer #2: Yes

3. Has the statistical analysis been performed appropriately and rigorously? 

Reviewer #1: Yes

Reviewer #2: Yes

4. Have the authors made all data underlying the findings in their manuscript fully available?

Reviewer #1: Yes

Reviewer #2: Yes

5. Is the manuscript presented in an intelligible fashion and written in standard English?

Reviewer #1: Yes

Reviewer #2: (No Response)

6. Review Comments to the Author

Reviewer #1: All my comments have been addressed. I loved reading and reviewing this manuscript. I hope this is just the beginning of a series of research output where authors can nicely merge highly technical model development skills and sport science practice.

Reviewer #2: Thank you for the opportunity to review t manuscript. The author examined novel kinematics and deep learning-based algorithm for the detection of step events in treadmill running. My all comments were revised.

7. PLOS authors have the option to publish the peer review history of their article (what does this mean?). If published, this will include your full peer review and any attached files.

Reviewer #1: **Yes: **Andrea Zignoli

Reviewer #2: No

---

## [Editor Report · Acceptance letter]

30 Jul 2021

PONE-D-21-06501R1 

Development and validation of FootNet; a new kinematic algorithm to improve foot-strike and toe-off detection in treadmill running 

Dear Dr. Rivadulla:

I'm pleased to inform you that your manuscript has been deemed suitable for publication in PLOS ONE. Congratulations! Your manuscript is now with our production department. 

Kind regards, 

on behalf of

Dr Laurent Mourot 

Academic Editor

PLOS ONE